# Morphological and Molecular Studies of Three New Diatom Species from Mountain Streams in South Korea

Eun-A Hwang [1], Ha-Kyung Kim [1], In-Hwan Cho [1], Chen Yi [1] and Baik-Ho Kim [1,2,*]

[1] Department of Environmental Science, Hanyang University, Seoul 04763, Korea
[2] Department of Life Science and Research Institute for Natural Sciences, Hanyang University, Seoul 04763, Korea
* Correspondence: tigerk@hanyang.ac.kr; Tel.: +82-2-2220-0960

**Abstract:** In January 2019, epilithic diatoms were collected from two streams on Mount Gumdan and Mount Yongma near Lake Paldang in central South Korea. A total of 16 diatoms were isolated and classified by molecular and morphological analysis. Morphology was studied by LM and SEM, while the molecular study focused on small subunit (SSU) rRNA and ribulose bisphosphate carboxylase (rbcL) genes. Molecular analysis showed that the three species had clear differences in phylogenetic distance. Based on these findings, we studied the ultrastructure of three species. Among the morphological characteristics, *Hannaea librata* is longer but narrower and always has conical spines, while the similar species *H. pamirensis* has bifurcated spines in the central region and conical spines near the pole. *Gomphonema seminulum* is wider in the axial–central area than *G. pumilium*. *Nitzschia inclinata* has a bended valve apex, while *N. oligotraphenta* has a straight apex.

**Keywords:** *Hannaea*; *Gomphonema*; *Nitzschia*; morphology; phylogeny tree; taxonomy

## 1. Introduction

Diatoms are one of the most diverse groups of microalgae in aquatic habitats, with over 30,000 described and many more undescribed species [1]. They inhabit most of the world and have characteristics that are regulated by environmental conditions such as preferred water temperature, nutrients, conductivity, pH, and salinity [2–5], so diatoms are useful as bioindicators for interpreting the aquatic environment [6]. They have siliceous cell walls with intricate ornamentation and perforations [7,8], and their taxonomy and systematics are based on the complex patterns of their cell walls [7,9]. Therefore, reliable identification provides a consistent basis for further advances in ecological research [10].

Until the 19th century, most studies of diatom morphology were conducted using light microscopy (LM) at 1000× magnification [11,12], but LM has limitations for understanding small species and ultrastructures. Therefore, recent protocols include observing ultrastructures using scanning electron microscopy (SEM). In addition, accurate identification is achieved through molecular analysis. Among the genes used in this analysis, the small subunit (SSU) rRNA coding gene is the most widely used and suitable for inferring phylogenetic relationships [13,14], and the ribulose-1,5-bisphosphate carboxylase/oxygenase large subunit (rbcL) gene appears more suitable for evolutionary study [14].

Lake Paldang, the main source of water for citizens of Seoul and the metropolitan area, is one element of the Han River reservoir cooperation system [15,16]. Human activity is accelerating water pollution, and it is necessary to monitor the water quality of rivers and mountain streams in this area [12]. Mountain streams have harsh environmental conditions, such as low water temperature, poor nutrient status, and high turbidity, unlike rivers and lakes. In mountain streams, diatoms are the most abundant and dominant producers and are important taxa that can dominate the entire aquatic ecosystem [17]. As a result, the species composition of diatoms that inhabit the diverse environments of mountain streams varies, but many species have not yet been studied.

Therefore, this study conducted a morphological and molecular analysis of diatoms collected from streams on Mt. Gumdan and Yongma, which flow into Lake Paldang. We identified three new diatom species.

## 2. Materials and Methods

*Sample collection*: In January 2019, the epilithic diatoms were collected from streams on Mt. Gumdan (37°30′35″ N 127°16′24″ E) and Mt. Yongma (37°29′46″ N 127°16′55″ E) near Lake Paldang in central Korea (Figure 1). There was thick ice on the water surface at the time of collection, and the sediments consisted mostly of bedrock and some gravel. Each sample was collected by scraping 25 cm$^2$ of the stone surface with a toothbrush.

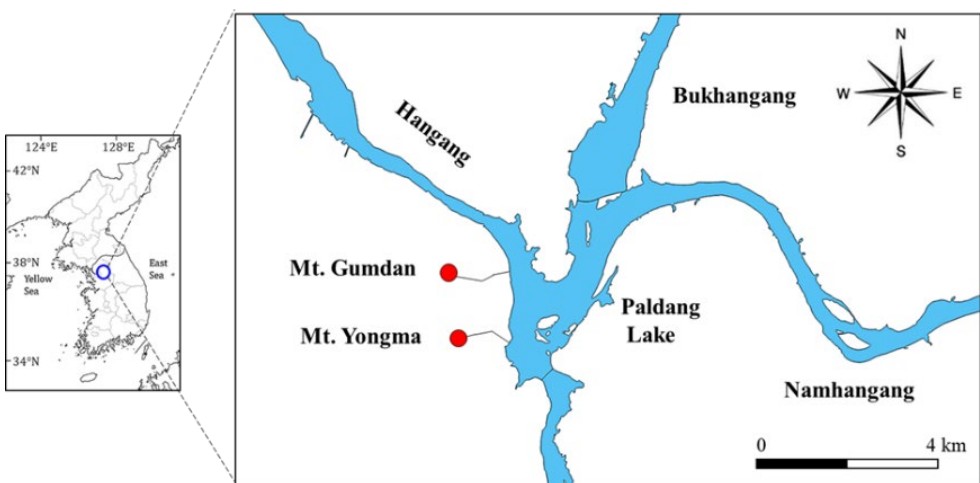

**Figure 1.** Map of sampling sites in mountain streams from Korea.

*Isolation*: Under an inverted microscope (Olympus, Tokyo, Japan), diatoms were isolated by the capillary method [18] using Pasteur pipettes (Hilgenberg GmbH, Malsfeld, Germany). To minimize contamination by bacteria, fungi, and other algae, sterilized diatom medium (DM) [19] was dropped on the glass slide, and each cell was washed and then isolated on 96-well plates containing 160 μL of DM in each well.

*Culture*: At 10–14 days after isolation, when cells had reached the exponential growth stage, they were transferred to 24-well plates containing 1 mL of DM. After 1 week, cells were transferred to 25 cm$^3$ flasks containing 20 mL DM. To maintain the health of the diatoms, the strains were subcultured at 30–50-day intervals. Samples were incubated at 20 °C under cool white fluorescent lamps with 12:12 h light:dark cycles and light intensity set to 30–50 μmol m$^{-2}$ s$^{-1}$ [12,20,21].

*Diatom preparation for permanent slides*: Morphological analysis—organic material was removed to observe morphological features. Each harvested subculture sample and acid were mixed at a 2:1 ratio (acid was a 1:1 mixture of sulfuric and nitric acid) and boiled for 2–3 minutes. To remove acid, we added distilled water, allowed samples to settle for 1 day, and carefully removed the supernatant with a pipette. This process was repeated 4–5 times per sample.

*Light microscope (LM) observation*: A few drops of pretreated sample were dropped onto cover glass and completely evaporated. Permanent slides were made using Mountain media (Wako Pure Chemical Industries, Ltd., Osaka, Japan) with a refraction index of greater than 1.5 for LM observation (Nikon E600, Tokyo, Japan) and photography (AmScope ToupView 3.7, Irvine, CA, USA).

*Scanning electron microscope (SEM) observation*: The pretreated sample was filtered using GTTP Millipore filter membrane (Millipore Filter Corporation, Cork, Ireland). The membrane was placed on an SEM stub with attached carbon tape (Shintron Enterprise Co., Ltd., Kaohsiung, Taiwan) and dried at room temperature for 24 h. Then, platinum coating

was applied for 120 s using coater (Hyun corporation, Seoul, Korea), and images were acquired by SEM (Thermo fisher scientific, Waltham, MA, USA).

*DNA extraction and PCR amplification*: We harvested cells from the cultured flask and transferred them to 1.5 mL microtubes for centrifugation at 4000 rpm for 10 min. DNA was extracted using the DNeasy Plant mini kit (Qiagen, Valencia, CA, USA). PCR reaction mixtures of 40 μL contained 23.8 μL of distilled water, 4 μL of 10x Ex PCR buffer (TaKaRa, Tokyo, Japan), 4 μL of dNTP (TakaRa), 0.2 μL of Ex Taq polymerase (TaKaRa), 4 μL of DNA extraction template, and 2 μL of each primer. PCR was used to amplify the small subunit ribosomal RNA (SSU rRNA) and ribulose-1,5-bisphosphate carboxylase/oxygenase (rbcL) genes, as shown in Table 1. PCR assays were conducted in a Bio-Rad iCycler (Bio-Rad, Hercules, CA, USA) as follows: predenaturation at 94 °C for 4 min, 37 cycles at 94 °C for 20 s, 56 °C for 30 s, and 72 °C for 50 s, and final extension at 72 °C for 5 min. To confirm the PCR results, electrophoresis (ADVANCE, Tokyo, Japan) was conducted at 120 v using 1% agarose gel with 1% staining solution (Genetics, Dueren, Germany).

**Table 1.** Primers used for amplification and sequencing of the nuclear SSU rRNA and rbcL.

| Gene | Primer | Nucleotide Sequence (5′ to 3′) | References |
|---|---|---|---|
| SSU rRNA | AT18F01 | YAC CTG GTT GAT CCT GCC AGT AG | [21] |
| | AT18R02 | GTTTCAGCCTTGCGACCATACTCC | [21] |
| | AT18F02 | AGA ACG AAA GTT AAG GGA TCG AAG ACG | [21] |
| | AT18R01 | GCTTGATCCTTCTGCAGGTTCACC | [21] |
| | EulA | AAC CTG GTT GAT CCT GCC AGT | [22] |
| | EukB | GAT CCT TCT GCA GGT TCA CCT AC | [22] |
| rbcL | F3 | GCT TAC CGT GTA GAT CCA GTT CC | [23] |
| | R3 | CCT TCT AAT TTA CCA ACA ACT G | [23] |

*Phylogenetic analysis*: DNA sequences were assembled using BioEdit v. 7.0.5.3 (Sequence Alignment Editor, Carlsbad, CA, USA, Hall 1999), and SSU rRNA and rbcL sequences were deposited in Genbank [22–24]. SSU and rbcL sequences of all species used in the phylogenetic analysis were taken from the National Center for Biotechnology Information (NCBI) and compared. Clustal W multiple alignment [25] was conducted in BioEdit v. 7.0.5.3 to match the sequence lengths of our and related species. MEGA version 7.0 [26] was used to calculate and represent phylogenetic relationships among the species. Phylogenetic trees were estimated by maximum likelihood based on the Kimura 2-parameter model [26]. Among the 24 models in MEGA 7.0, GTR + G + I was selected as the most appropriate. Bootstrap support was calculated with 1000 replicates for each branch of the phylogenetic tree. To calculate the similarity score and genetic distance (*p*-distance), 1000 bootstrap replicates and the Kimura 2-parameter model were also used in BioEdit v. 7.0.5.3 and MEGA 7.0.

## 3. Results and Discussion

*3.1. Morphological Characteristics of Hannaea librata sp. nov.*

Class Bacillariophyceae
Subclass Fragilariophycidae
Order Licmophorales
Family Ulnariaceae
Genus *Hannaea*
*Hannaea librata* E.A. Hwang and B.H. Kim (Figure 2, LM; Figure 3, SEM)

**Description:** Valves are linear with strongly rostrate apices, slightly arcuate, and slight swelling at the central area of the curved (ventral) side. Length 44–99 μm, width 5–5.5 μm, 12–16 striae in 10 μm, and 70–80 areolae in 10 μm. Axial area is narrow, linear, and slightly bent in the central region. Striae are alternate, uniseriate, perpendicular to the axial area, and parallel to each other, but one or two striae at the apical end were parallel to the axis. Apical pore field is located at each pole of the valve. Valve face is undulating; valve with

striae is slightly sunken, and without striae is swollen. In central area, absent striae but the undulating valve face forms ghost striae and forms a wide U-shape extending into both striae. Valve mantle is flat regardless of striae. Areolae are poroid type, elongated oval in shape, and near the axial and margin are round in shape. Single rimoportula with slit-like opening located at valve apices. Girdle band is open ring type with single row of round areolae and scalloped advalvar edge. Spines present along the valve face–mantle junction until apical, shape conical in all parts of the valve, and tips radiate from the central area. In girdle view, frustules are rectangular. Cells form linear colonies.

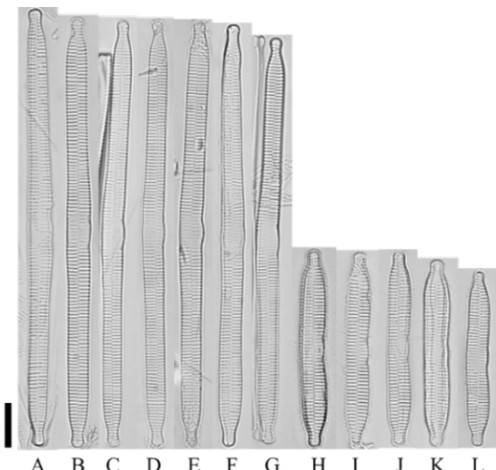

**Figure 2.** Light microscope photo of *Hannaea librata* sp. nov. (**A–L**) valve view. (**A–G**) taken from isotype population (AG9002); (**H–L**) from holotype population (AG001); scale bar = 10 μm.

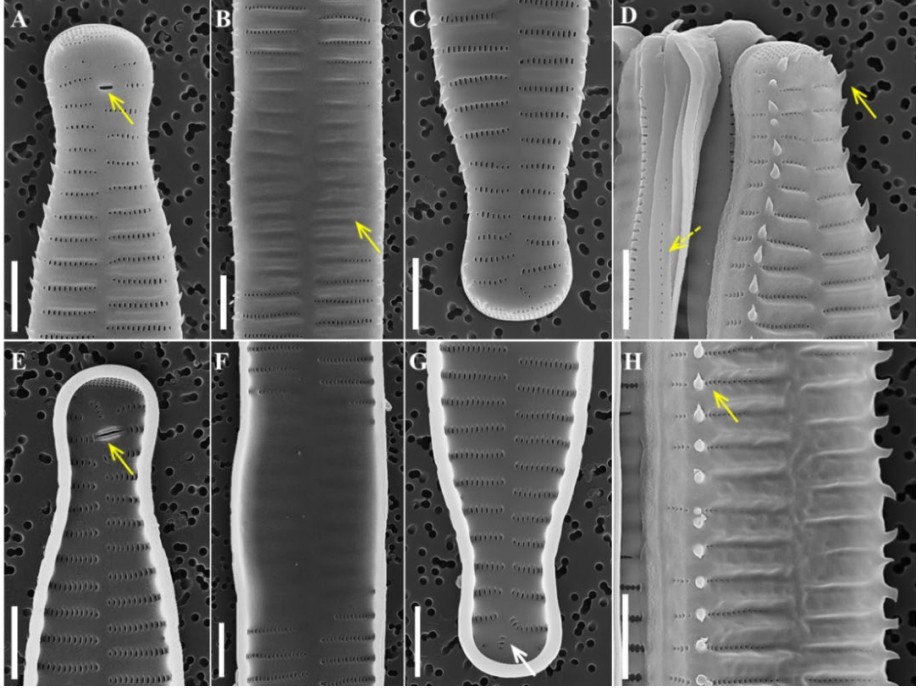

**Figure 3.** SEM of *Hannaea librata* sp. nov. taken from holotype population (AG9001), scale bar = 2 μm: (**A**) external view of apex with rimoportula (arrow); (**B**) external view of central area, ghost strain (arrow); (**C**) external view without rimoportula; (**D**) external view of one side showing valve apices with spines, conical shape (solid arrow), girdle band (dotted arrow); (**E**) internal view of apex with rimoportula (arrow); (**F**) internal view of central area; (**G**) internal view of apex without rimoportula, vertical striae (arrow); (**H**) external view of central area with spines, conical shape (arrow).

**Remarks:** To compare the morphological characteristics of *Hannaea librata* with similar species, its structure was observed using LM, and its ultrastructure was closely observed using SEM. The morphological characteristics of *H. librata* are different from those of related taxa (Table 2). The valve of *H. pamirensis* [27] is shorter and wider than that of *H. librata*, and the density of striae is higher. The central area of *H. librata* is wide and U-shaped, extending into both striae and wider than *H. pamirensis*. Moreover, the spine of *H. librata* is conical, but *H. pamirensis* has conical spines near each pole and bifurcated thorn-shaped spines near the center of the valve, showing a distinct difference. *H. hattoriana* [28] is another species with similar morphological characteristics and has a more lanceolate valve shape than *H. librata.*; *H hattoriana* has a narrower central area than *H. librata*. The valve shape of *H. recta* [29,30] is more lanceolate than *H. librata.* The valve mantle of *H. librata* is flat, while that of *H. recta* is waved. Lastly, the shape of the spine of *H. librata* is constant, whereas the shape of the *H. recta* changes from the center valve and each pole. *Hannaea* shares morphological similarities with both *Fragilaria* and *Synedra* [31] and is most closely related to *Fragilaria* [32]. In comparison with *F. capucina*, which has the most similar morphology, the shape spine in the center of the valve is conical in *H. librata* and triangular in *F. capucina*. In the center of the valve, *H. librata* has unilateral swelling, but *F. capucina* remains flat from both sides [33–35]. *Hannaea librata* is distinguished from similar related taxa in the shape of the valve, central area, and spine.

**Table 2.** Comparison of morphological characteristics for *Hannaea librata* nov. and closely related species.

| | *Hannaea librata* | *H. pamirensis* | *H. hattoriana* | *H. recta* | *Fragilaria capucina* |
|---|---|---|---|---|---|
| Length (µm) | 44–99 | 42–45 | 40–85 | 29–71 | 28–47 |
| Width (µm) | 5–5.5 | 5.5–7.0 | 6–7 | 6–7 | 3.3–4.2 |
| No. striae (/10 µm) | 12–16 | 17–18 | 13–14 | 12–14 | 14–17 |
| No. areolae (/10 µm) | 70–80 | 75–80 | n.d. | n.d. | n.d. |
| Valves | Linear | Weakly arcuate to almost linear | Lanceolate, slightly arcuate to almost linear | Lanceolate | Linear |
| Valve apices | Strongly rostrate | Capitate to rostrate apices | Capitate to rostrate apices | Rostrate | Weakly rostrate |
| Striation | Parallel | Parallel | Parallel | Parallel | |
| Ghost strain | Central area | Central area | Central area | Central area | n.d. |
| Valve face | Waved | Waved | n.d. | Waved | Alternate, parallel to slightly radiate toward the apices |
| Valve mantle | Flat | Flat | n.d. | Waved | n.d. |
| Valves of central area | Expanded to unilateral | Expanded to unilateral | Expanded to unilateral | Expanded to unilateral | Flat |
| Central area | Wide U-shape extending into both striae | Unilaterally tumid on the concave margin | Unilaterally tumid on the concave margin | Horseshoe shaped | Rectangular to rhombic |
| Girdle band | Single row of small punctuate | n.d. | n.d. | n.d. | n.d. |
| Apical pore fields | Rectangular | Rectangular | n.d. | Rectangular | n.d. |
| Spine shape | Conical | Bifurcated thorn (center)/ conical (near the pole) | n.d. | Bifurcated thorn (center)/ conical (near the pole) | Conical near the apex to triangular in the middle |
| Rimoportula | One per valve | One per valve | n.d. | One per valve | Two per valve |
| References | This study | [27] | [27,28] | [29,30] | [31–35] |

*Holotype:* permanent slides were deposited with the Korea Collection for Type Cultures KCTC under the deposit number AG9001.

*Isotype:* permanent slides were deposited with the Korea Collection for Type Cultures KCTC under the deposit numbers AG9002, AG9003, AG9006, and AG9007.

*Habitat:* free-living on rocky substrates. Water quality: WT (water temperature), 1.0 °C; DO (dissolved oxygen), 15.6 mg/L; pH, 6.4; EC (electrical conductivity), 42 μS/cm; turbidity, 8.5 NTU. Major species (more than 5% of the total abundance) recorded from the same location: *Achnanthes convergens*, *Gomphonema parvulum*, *Achnanthes minutissima*, *Fragilaria gracilis*, and *Gomphonema gracile*.

*Type locality:* Republic of Korea, Gyeonggi-do Province, Mount Gumdan, 37°30′35″ N 127°16′24″ E, 3 January 2019.

*Molecular characterization:* SSU, rbcL accession no. The nucleotide sequences of the SSU rRNA and rbcL genes of the strain were deposited in GenBank (NCBI) under the SSU accession numbers ON040635, ON040636, ON040640, ON040641, and ON040642, and rbcL accession numbers OP137173, OP183209, OP183212, OP183213, and OP183214.

*Etymology:* "librata" is derived from the Latin "librátĭo", which means that spines at the center and at each apex are constant in shape.

### 3.2. Phylogenetic Characteristics of H. librata sp. nov.

We selected several species that were similar molecularly to *Hannaea* using GenBank and inferred phylogenetic characteristics using 28 aligned sequences of SSU and 23 aligned sequences of rbcL (Figures 4 and 5). *Hannea* has been separated from *Ceratoneis*, and there are 20 species worldwide. However, there are few molecular phylogenetic studies, and the sequence is unknown. In the SSU and rbcL tree, *H. librata* collected in different places belong to the same clade. Its sister groups include *Fragilaria*, *Ulnaria*, and *Synedra*, which are distinguished from *Hannaea*. Genetic distance and similarity analysis results for SSU and rbcL indicated that the genetic distance between *H. librata* collected from different places was the same for all five individuals (Tables 3 and 4). *H. librata* showed a genetic distance of 0.015 or less from *Fragilaria* in SSU genes and 0.081 or less for the rbcL gene and appeared molecularly similar to *F. capucina*, with a genetic distance for the SSU gene of 0.011 and 0.012 for the rbcL gene, making it clearly distinct from *H. librata*. Therefore, the phylogenetic features of *H. librata* differentiate it from *Fragilaria*, occupying distinctly different clades, and are clearly distinguished from those of *F. capucina*, which is molecularly and phylogenetically similar.

**Table 3.** Similarity scores and genetic distances among 15 aligned sequences (1478 bp) based on SSU rRNA gene.

| Species | Accession | 1 | 2 | 3 | 4 | 5 | 6 | 7 | 8 | 9 | 10 | 11 | 12 | 13 | 14 | 15 |
|---|---|---|---|---|---|---|---|---|---|---|---|---|---|---|---|---|
| | | Similarity | | | | | | | | | | | | | | |
| **1** *Hannaea librata* | ON040635 | | 1.000 | 1.000 | 1.000 | 1.000 | 0.989 | 0.989 | 0.989 | 0.987 | 0.987 | 0.985 | 0.974 | 0.974 | 0.969 | 0.969 |
| **2** *Hannaea librata* | ON040636 | 0.000 | | 1.000 | 1.000 | 1.000 | 0.989 | 0.989 | 0.989 | 0.987 | 0.987 | 0.985 | 0.974 | 0.974 | 0.969 | 0.969 |
| **3** *Hannaea librata* | ON040640 | 0.000 | 0.000 | | 1.000 | 1.000 | 0.989 | 0.989 | 0.989 | 0.987 | 0.987 | 0.985 | 0.974 | 0.974 | 0.969 | 0.969 |
| **4** *Hannaea librata* | ON040641 | 0.000 | 0.000 | 0.000 | | 1.000 | 0.989 | 0.989 | 0.989 | 0.987 | 0.987 | 0.985 | 0.974 | 0.974 | 0.969 | 0.969 |
| **5** *Hannaea librata* | ON040642 | 0.000 | 0.000 | 0.000 | 0.000 | | 0.989 | 0.989 | 0.989 | 0.987 | 0.987 | 0.985 | 0.974 | 0.974 | 0.969 | 0.969 |
| **6** *Fragilaria capucina* | MH356257 | 0.011 | 0.011 | 0.011 | 0.011 | 0.011 | | 0.999 | 1.000 | 0.996 | 0.996 | 0.994 | 0.979 | 0.979 | 0.965 | 0.965 |
| **7** *F. capucina* **var.** *mesolepta* | MH997845 | 0.011 | 0.011 | 0.011 | 0.011 | 0.011 | 0.001 | | 0.999 | 0.996 | 0.996 | 0.992 | 0.979 | 0.979 | 0.966 | 0.966 |
| **8** *Fragilaria bidens* | AB430599 | 0.011 | 0.011 | 0.011 | 0.011 | 0.011 | 0.000 | 0.001 | | 0.996 | 0.996 | 0.994 | 0.979 | 0.979 | 0.965 | 0.965 |
| **9** *Fragilaria crotonensis* | AM712616 | 0.013 | 0.013 | 0.013 | 0.013 | 0.013 | 0.004 | 0.004 | 0.004 | | 1.000 | 0.997 | 0.977 | 0.977 | 0.965 | 0.965 |
| **10** *Fragilaria vaucheriae* | AM497733 | 0.013 | 0.013 | 0.013 | 0.013 | 0.013 | 0.004 | 0.004 | 0.004 | 0.000 | | 0.997 | 0.977 | 0.977 | 0.965 | 0.965 |
| **11** *Fragilaria vaucheriae* | EU260469 | 0.015 | 0.015 | 0.015 | 0.015 | 0.015 | 0.006 | 0.007 | 0.006 | 0.003 | 0.003 | | 0.974 | 0.974 | 0.963 | 0.963 |
| **12** *Synedra minuscula* | EF423415 | 0.027 | 0.027 | 0.027 | 0.027 | 0.027 | 0.022 | 0.022 | 0.022 | 0.024 | 0.024 | 0.026 | | 1.000 | 0.959 | 0.959 |
| **13** *Synedra hyperborea* | AY485464 | 0.027 | 0.027 | 0.027 | 0.027 | 0.027 | 0.022 | 0.022 | 0.022 | 0.024 | 0.024 | 0.026 | 0.001 | | 0.959 | 0.959 |
| **14** *Ulnaria acus* | KF959659 | 0.030 | 0.030 | 0.030 | 0.030 | 0.030 | 0.034 | 0.033 | 0.034 | 0.034 | 0.034 | 0.036 | 0.041 | 0.041 | | 0.999 |
| **15** *Ulnaria ulna* | MG684361 | 0.030 | 0.030 | 0.030 | 0.030 | 0.030 | 0.034 | 0.033 | 0.034 | 0.034 | 0.034 | 0.036 | 0.041 | 0.041 | 0.001 | |
| | | *p*-distance | | | | | | | | | | | | | | |

Accession; Gene Bank accession number.

**Table 4.** Similarity scores and genetic distances among 17 aligned sequences (596 bp) based on rbcL gene.

| Species | Accession | 1 | 2 | 3 | 4 | 5 | 6 | 7 | 8 | 9 | 10 | 11 | 12 | 13 | 14 | 15 | 16 | 17 |
|---|---|---|---|---|---|---|---|---|---|---|---|---|---|---|---|---|---|---|
| | | Similarity | | | | | | | | | | | | | | | | |
| **1** *Hannaea librata* | OP137173 | | 1.000 | 1.000 | 1.000 | 1.000 | 0.988 | 0.980 | 0.986 | 0.982 | 0.984 | 0.984 | 0.982 | 0.914 | 0.914 | 0.947 | 0.945 | 0.947 |
| **2** *Hannaea librata* | OP183209 | 0.000 | | 1.000 | 1.000 | 1.000 | 0.988 | 0.980 | 0.986 | 0.982 | 0.984 | 0.984 | 0.982 | 0.914 | 0.914 | 0.947 | 0.945 | 0.947 |
| **3** *Hannaea librata* | OP183212 | 0.000 | 0.000 | | 1.000 | 1.000 | 0.988 | 0.980 | 0.986 | 0.982 | 0.984 | 0.984 | 0.982 | 0.914 | 0.914 | 0.947 | 0.945 | 0.947 |
| **4** *Hannaea librata* | OP183213 | 0.000 | 0.000 | 0.000 | | 1.000 | 0.988 | 0.980 | 0.986 | 0.982 | 0.984 | 0.984 | 0.982 | 0.914 | 0.914 | 0.947 | 0.945 | 0.947 |
| **5** *Hannaea librata* | OP183214 | 0.000 | 0.000 | 0.000 | 0.000 | | 0.988 | 0.980 | 0.986 | 0.982 | 0.984 | 0.984 | 0.982 | 0.914 | 0.914 | 0.947 | 0.945 | 0.947 |
| **6** *Fragilaria capucina* | KT072928 | 0.012 | 0.012 | 0.012 | 0.012 | 0.012 | | 0.986 | 0.998 | 0.994 | 0.996 | 0.990 | 0.988 | 0.921 | 0.921 | 0.945 | 0.943 | 0.945 |
| **7** *Fragilaria capucina* | KC736594 | 0.020 | 0.020 | 0.020 | 0.020 | 0.020 | 0.014 | | 0.988 | 0.984 | 0.986 | 0.988 | 0.986 | 0.921 | 0.921 | 0.945 | 0.943 | 0.945 |
| **8** *Fragilaria bidens* | AB430676 | 0.014 | 0.014 | 0.014 | 0.014 | 0.014 | 0.002 | 0.012 | | 0.992 | 0.994 | 0.992 | 0.990 | 0.923 | 0.923 | 0.947 | 0.945 | 0.947 |
| **9** *Fragilaria crotonensis* | KF959640 | 0.018 | 0.018 | 0.018 | 0.018 | 0.018 | 0.006 | 0.016 | 0.008 | | 0.998 | 0.988 | 0.986 | 0.917 | 0.917 | 0.941 | 0.938 | 0.941 |
| **10** *Fragilaria crotonensis* | KT072903 | 0.016 | 0.016 | 0.016 | 0.016 | 0.016 | 0.004 | 0.014 | 0.006 | 0.002 | | 0.990 | 0.988 | 0.919 | 0.919 | 0.943 | 0.941 | 0.943 |
| **11** *Fragilaria vaucheriae* | HYUD010 | 0.016 | 0.016 | 0.016 | 0.016 | 0.016 | 0.010 | 0.012 | 0.008 | 0.012 | 0.010 | | 0.994 | 0.923 | 0.923 | 0.947 | 0.945 | 0.947 |
| **12** *Fragilaria perminuta* | KF959650 | 0.018 | 0.018 | 0.018 | 0.018 | 0.018 | 0.012 | 0.014 | 0.010 | 0.014 | 0.012 | 0.006 | | 0.921 | 0.921 | 0.945 | 0.943 | 0.945 |
| **13** *Synedra minuscula* | JN162825 | 0.081 | 0.081 | 0.081 | 0.081 | 0.081 | 0.075 | 0.075 | 0.073 | 0.079 | 0.077 | 0.073 | 0.075 | | 1.000 | 0.934 | 0.936 | 0.934 |
| **14** *Synedra hyperborea* | HQ912485 | 0.081 | 0.081 | 0.081 | 0.081 | 0.081 | 0.075 | 0.075 | 0.073 | 0.079 | 0.077 | 0.073 | 0.075 | 0.000 | | 0.934 | 0.936 | 0.934 |
| **15** *Ulnaria acus* | KF959645 | 0.051 | 0.051 | 0.051 | 0.051 | 0.051 | 0.053 | 0.053 | 0.051 | 0.057 | 0.055 | 0.051 | 0.053 | 0.063 | 0.063 | | 0.998 | 1.000 |
| **16** *Ulnaria ulna* | MG684332 | 0.053 | 0.053 | 0.053 | 0.053 | 0.053 | 0.055 | 0.055 | 0.053 | 0.059 | 0.057 | 0.053 | 0.055 | 0.061 | 0.061 | 0.002 | | 0.998 |
| **17** *Ulnaria ulna* | KT072942 | 0.051 | 0.051 | 0.051 | 0.051 | 0.051 | 0.053 | 0.053 | 0.051 | 0.057 | 0.055 | 0.051 | 0.053 | 0.063 | 0.063 | 0.000 | 0.002 | |
| | | *p*-distance | | | | | | | | | | | | | | | | |

Accession: GenBank accession number.

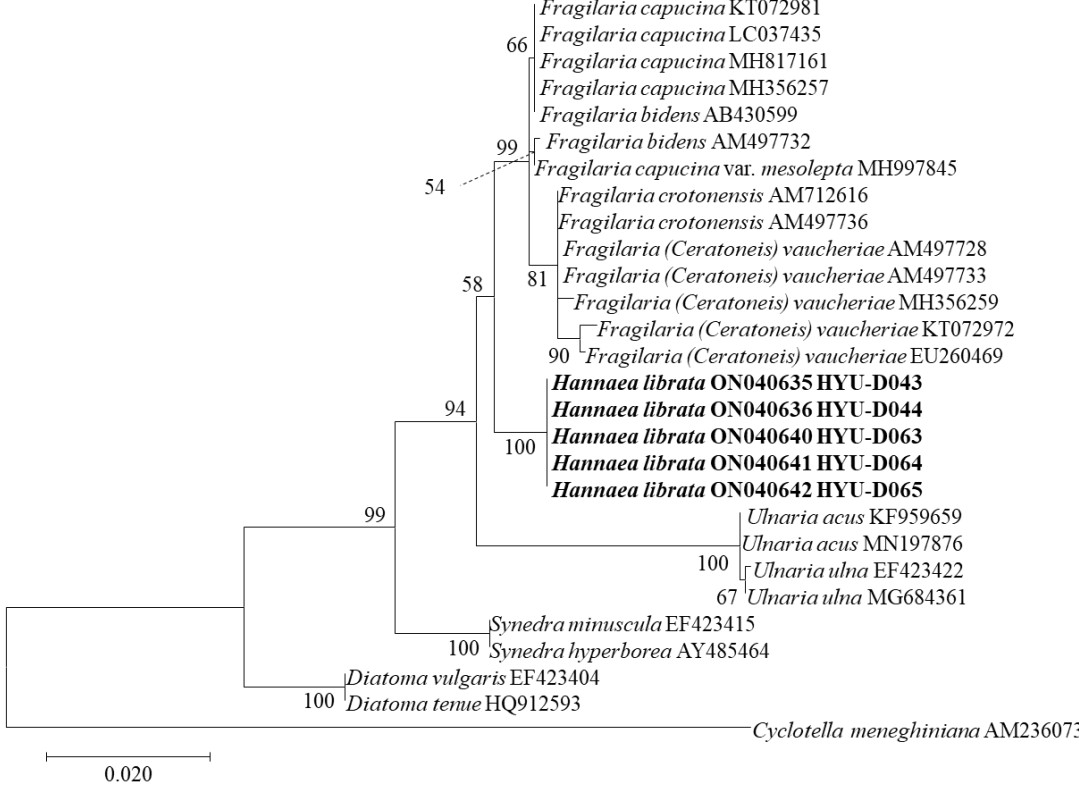

**Figure 4.** SSU rRNA maximum likelihood phylogenetic tree of *Hannaea librata* sp. nov. and other species. The tree with the highest log likelihood (−3434.22) is shown. The percentages of trees in which the associated taxa clustered together are shown next to the branches. The analysis included 28 nucleotide sequences and 1478 positions in the final data set. *Cyclotella meneghiniana* was used as an outgroup.

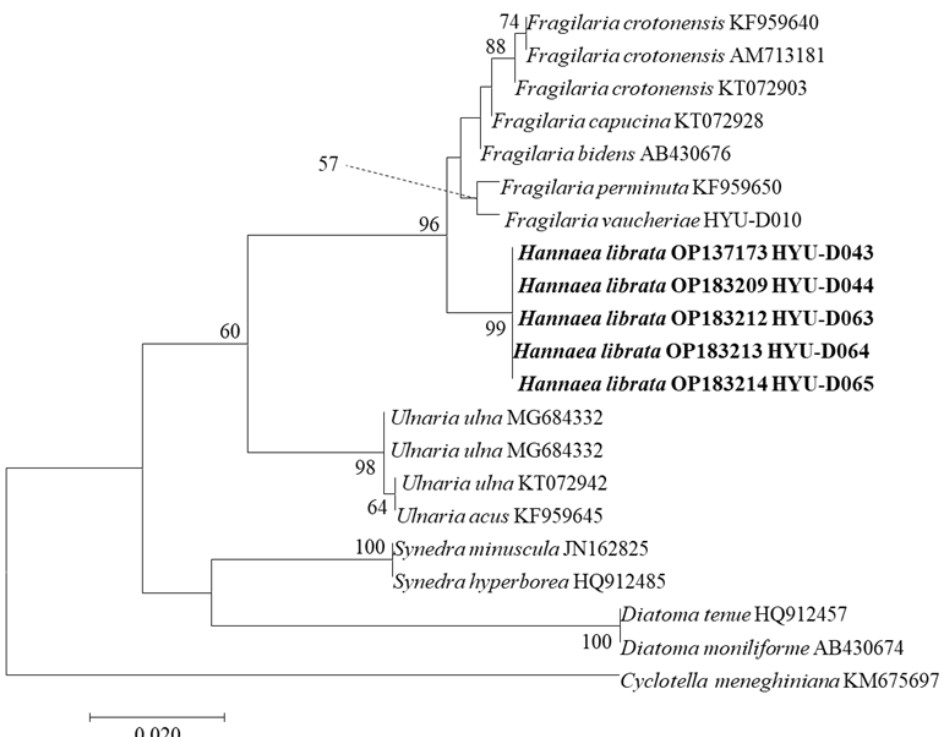

**Figure 5.** rbcL maximum likelihood phylogenetic tree of *Hannaea librata* sp. nov. and other species. The tree with the highest log likelihood (−1639.58) is shown. The percentages of trees in which the associated taxa clustered together are shown next to the branches. The analysis involved 22 nucleotide sequences and 596 positions in the final data set. *Cyclotella meneghiniana* was used as the outgroup.

*3.3. Morphological Characteristics of Gomphonema Seminulum* sp. nov.

 Class Bacillariophyceae
 Subclass Bacillariophycidae
 Order Cymbellales
 Family Gomphonemataceae
 Genus *Gomphonema*
 *Gomphonema seminulum* E.A. Hwang and B.H. Kim (Figure 6, LM; Figure 7, SEM)

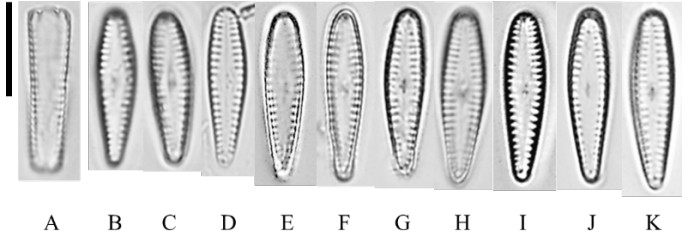

**Figure 6.** Light microscope photo of *Gomphonema seminulum* sp. nov. taken from holotype population (AG9005): (**A**) girdle view; (**B–K**) valve view; scale bar = 10 μm.

 **Description:** Valves are narrow lanceolate, slightly asymmetric about the axis, valve apices are bluntly rounded, and central and axial area is wide lanceolate shape. Length 15–20 μm, width 4–5 μm, 13–15 striae in 10 μm, and 40–50 areolae in 10 μm. Striae are uniseriate and parallel to slightly radiate toward the apices. Areola is broadly C-shaped, occluded by raised flaps; 2–4 lines from the axis are flapped open toward the center, and the other lines are open in the opposite direction. The large openings of the alveolus are formed inside the valve, and the areola is located outside the valve along this ultrastructure. Single stigma is circular in shape, located at the center of valve. Raphe extends along the

entire valvar face and is slightly undulate. External raphe ending, with central endings terminating in elliptical central pores, and polar endings deflecting in the opposite direction of the stigma. Internal raphe ending, with central endings bent to stigma and hook-shaped toward each pole, and polar endings forming helictoglossa. The apical pore field is located at the foot pole of the valve and is bifurcated by the raphe. Pseudosepta is formed at each apex of the internal valve. In girdle view, frustules are wedge-shaped.

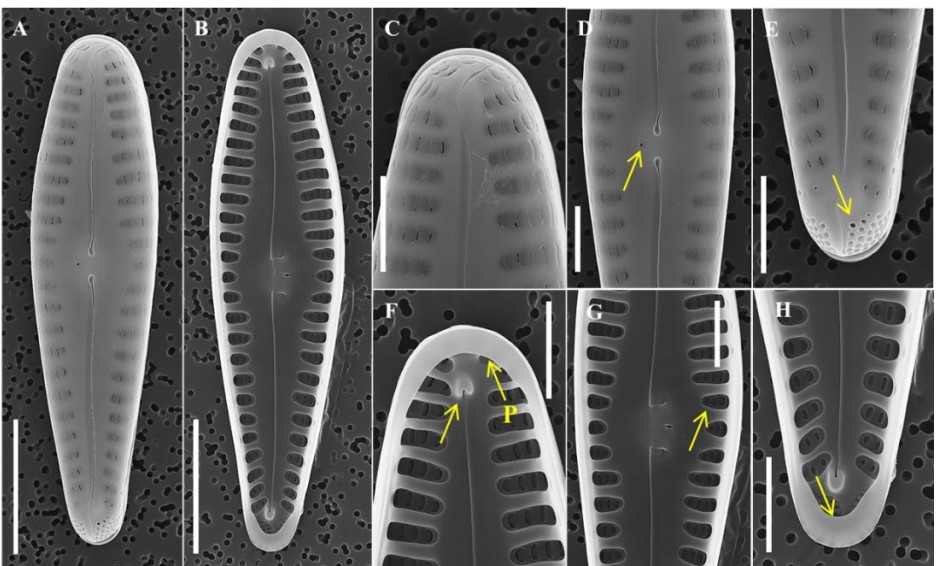

**Figure 7.** SEM of *Gomphonema seminulum* sp. nov. taken from holotype population (AG9005): (**A**,**B**) scale bar = 5 µm; (**C**–**H**) scale bar = 2 µm; (**A**) external view of whole valve; (**B**) internal view of whole valve; (**C**) external view of head pole; (**D**) external view of valve middle, showing the stigma on central area of valve face (arrow); (**E**) internal view of foot pole, showing the apical pore field (arrow); (**F**) internal view of head pole, showing the helictoglossa (arrow), pseudosepta (arrow P); (**G**) internal view of valve middle, showing the alveoli (arrow); (**H**) internal view of foot pole, showing the pseudosepta (arrow).

**Remarks:** In this study, we conducted LM and SEM to study the morphology of *G. seminulum* and confirmed new species belonging to the genus *Gomphonema*. The morphology of *G. seminulum* differs from closely related taxa (Table 5); among them, *G. bourbonense* [36], which lacks descriptions of ultrastructure, was further analyzed by examining type material (specimen ID TCC441, TCC458, TCC930 in the Thonon Culture Collection, France). The *G. seminulum* valve is narrower and more lanceolate than that of *G. bourbonense*. In addition, the valve of *G. seminulum* is asymmetric with a very wide central area, whereas *G. bourbonense* is symmetric with a narrower central area. Finally, C-shaped areolae in *G. seminulum* are wider than in *G. bourbonense*. It shows distinct differences from *G. pumilum*, another species that is morphologically similar to *G. seminulum*. *G pumilum* striae slightly radiate to strongly radiate toward the apices, but *G. seminulum* is almost parallel in the central area, slightly radiate near the head pole, and strongly radiate toward the foot pole. The valve of the head pole in *G. pumilum* is strongly tapered compared with *G. seminulum*. *G angustum* has a significantly lower central area density of striae than *G. seminulum*. Lastly, in *G. clevei*, the valve is rhombic or elliptical clavate, whereas *G. seminulum* is narrow lanceolate, and the external polar raphe ending of *G. clevei* is curved toward the stigma but deflected in the opposite direction in *G. seminulum*. Based on these observations, *G. seminulum* is clearly distinguished from closely related taxa.

**Table 5.** Comparison of morphological characteristics for *Gomphonema seminulum* sp. nov. and closely related species.

| | *Gomphonema seminulum* | *G. bourbonense* | *G. pumilum* | *G. angustum* | *G. clevei* |
|---|---|---|---|---|---|
| Length (μm) | 15–20 | 9.4–28 | 17–37 | 13–130 | 11.5–34 |
| Width (μm) | 4–5 | 3.3–4.7 | 5–8 | 3–12 | 3.5–7 |
| No. striae (/10 μm) | 13–15 | 10.5–13 | 11–14 | 11–15 | 10–15 |
| No. areolae (/1 μm) | 4–5 | n.d. | n.d. | n.d. | 3–4 |
| Striae | Parallel to slightly radiate near the head pole, strongly radiate toward the foot pole | Slightly radiate, almost parallel | Slight radiate to strongly radiate toward the apices | Slightly radiate, parallel, low density of central area | Slightly radiate |
| Areola | Broadly C-shaped, occluded by raised flaps, 2–4 lines from the axis are flap open toward the center, and the other lines are open in the opposite direction | Occluded by raised flaps, C-shaped, slit-like | C-shaped, 1–3 lines from the axis are flap open toward the center, and the other lines are open in the opposite direction | n.d. | C-shaped, occluded by raised flaps, 2–3 lines from the axis are flap open toward the center, and the other lines are open in the opposite direction |
| Axial area | Wide lanceolate | Anguste lanceolate | Small | Wide rectangle | Wide lanceolate |
| Stigma | One per valve, circular | One per valve, circular | One per valve | One per valve | One per valve, circular with a thickened margin |
| Valves | Narrow lanceolate, slightly asymmetric about the axis | Linear and slightly elliptical | Narrow elliptic lanceolate, tapering more strongly toward the head pole | Narrow lanceolate–ovata | Rhombic clavate, elliptical clavate |
| Valve apices | Bluntly rounded | Obtuse, wide circular | bluntly rounded, slightly protracted apex | Wide circular | Obtuse (head pole is wider than foot pole) |
| Raphe | Extends along the entire valvar face, slightly undulate | Parallel | Extends along the entire valvar face | Extends along the entire valvar face | Undulate (external), relatively linear (internal) |
| Central raphe ending | Elliptical (external), bent to stigma, hooks in opposite directions (internal) | Elliptical(external), bent to stigma, hooks in opposite directions (internal) | Elliptical (external), sharp hook shape (internal) | Hook shape | Elliptical (external), bending to stigma, hook shape (internal) |
| Polar raphe ending | deflects in the opposite direction of the stigma (external), formed helictoglossa (internal) | n.d. | Deflects in the opposite direction of the stigma (external), formed helictoglossa (internal) | n.d. | Curved toward the stigma (external), formed small helictoglossa (internal) |
| Reference | This study | [36] | [36–38] | [37–39] | [30,40] |

*Holotype:* permanent slides were deposited with the Korea Collection for Type Cultures KCTC under the deposit number AG9005.

*Habitat:* free-living on rocky substrates. Water quality: WT, 1.3 °C; DO, 14.6 mg/L; pH, 5.5; EC, 49 μS/cm; turbidity, 10.7 NTU. Major species (more than 5% of the total abundance) recorded from the same location: *Gomphoneis quadripunctata*, *Meridion constrictum*, *Achnanthidium minutissimum*, and *Achnanthes convergens*.

*Type locality:* Republic of Korea, Gyeonggi-do Province, Mount Yongma, 37°29′46″ N 127°16′55″ E, 3 January 2019.

*Molecular characterization:* SSU, rbcL accession no. The nucleotide sequences of the SSU rRNA and rbcL genes of the strain were deposited in GenBank (NCBI) under the accession numbers ON040637 and OP183210.

*Etymology:* "*seminulu*" is derived from the Latin "seminúdus", which means that the central area of *G. seminulum* is empty and simple.

### 3.4. Phylogenetic Characteristics of Gomphonema Seminulum sp. nov.

We used 20 aligned sequences of SSU rRNA and 18 aligned sequences of rbcL from GenBank and inferred the phylogenetic characteristics of species (Figures 8 and 9). In the SSU rRNA and rbcL trees, *G. seminulum* was placed in the *Gomphonema* clade and was distant from other similar genera. In the *Gomphonema* clade, *G. seminulum* was closer to the clade in which *G. bourbonense*, *G. pumilum*, *G. truncatum*, *G. subclavatum*, and *G. acuminatum* are placed, and its sister taxon was *G. bourbonense*. However, they were not close to *G. parvulum*, *G. affine* and *G. clevei*. Among the SSU rRNA alignment sequences (Table 6), *G. bourbonense* showed the smallest genetic distance from *G. seminulum* at 0.007. However, among morphologically similar species, *G. pumilum* was 0.008, and *G. clevei* was 0.031, which were greater distances. In rbcL-aligned sequences (Table 7), *G. acuminatum* showed the smallest genetic distance from *G. seminulum* of 0.037, while *G. bourbonense*, located in a nearby clade, was 0.044, and *G. pumilum* was 0.049. In the SSU rRNA and rbcL phylogenetic analysis, *G. seminulum* and *G. bourbonense* were close genetically and located in the neighboring clades, but the molecular phylogenetic characteristics of the two species were distinct. In addition, within the *Gomponema* clade, morphologically similar species such as *G. clevei* were clearly distinguished in a different clade. As a result, morphological and phylogenetic characteristics of *G. seminulum* clearly distinguish it from similar species.

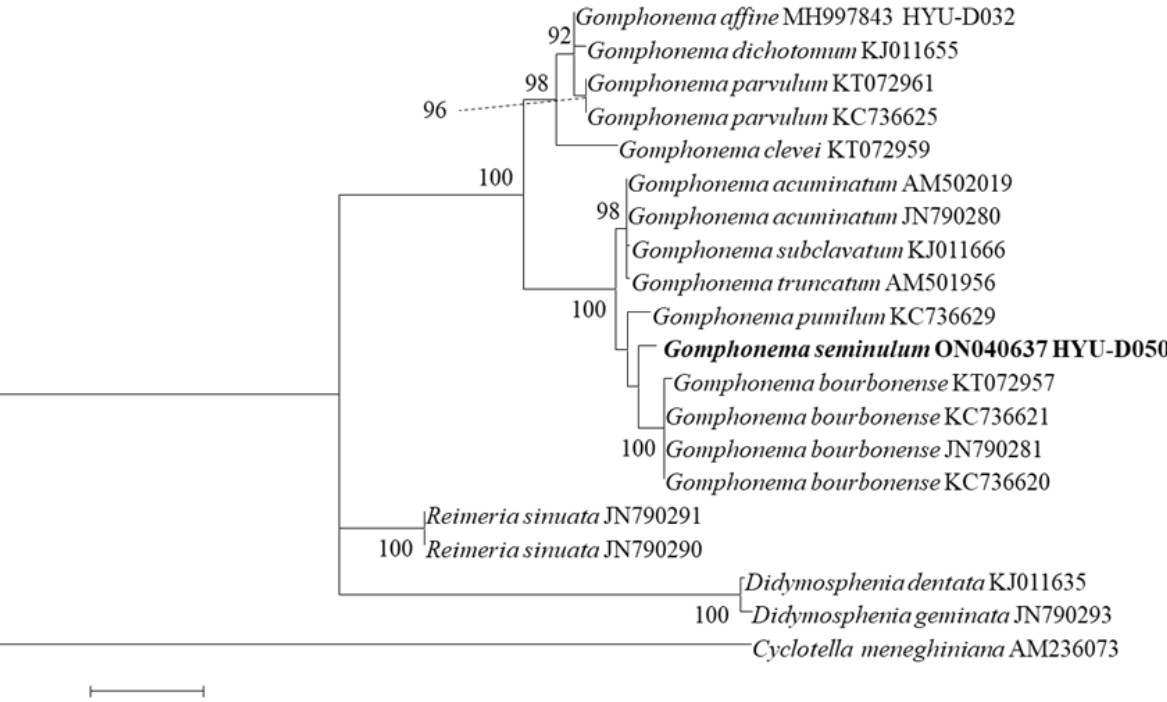

**Figure 8.** SSU rRNA phylogenetic tree by maximum likelihood method of *Gomphonema seminulum* sp. nov. and other species molecular position. The tree with the highest log likelihood (−3766.04) is shown. The percentages of trees in which the associated taxa clustered together are shown next to the branches. The analysis involved 20 nucleotide sequences and 1475 positions in the final data set. *Cyclotella meneghiniana* was used as the outgroup.

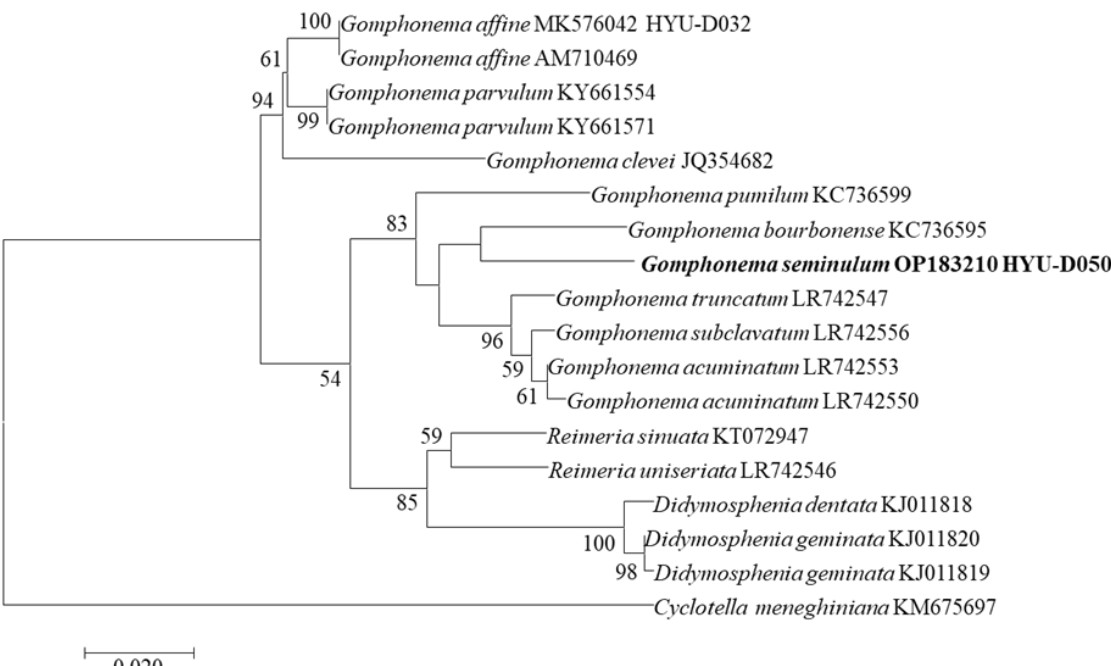

**Figure 9.** rbcL phylogenetic tree by maximum likelihood method of *Gomphonema seminulum* sp. nov. and other species molecular position. The tree with the highest log likelihood (−1964.35) is shown. The percentages of trees in which the associated taxa clustered together are shown next to the branches. The analysis involved 18 nucleotide sequences and 596 positions in the final data set. *Cyclotella meneghiniana* was used as the outgroup.

**Table 6.** Similarity scores and genetic distances among 15 aligned sequences (1475 bp) of *Gomphonema* species based on SSU rRNA gene.

| Species | Accession | 1 | 2 | 3 | 4 | 5 | 6 | 7 | 8 | 9 | 10 | 11 | 12 | 13 | 14 | 15 |
|---|---|---|---|---|---|---|---|---|---|---|---|---|---|---|---|---|
| | | Similarity | | | | | | | | | | | | | | |
| **1** *Gomphonema seminulum* | ON040637 | | 0.992 | 0.992 | 0.992 | 0.991 | 0.992 | 0.992 | 0.992 | 0.991 | 0.991 | 0.976 | 0.976 | 0.974 | 0.974 | 0.969 |
| **2** *Gomphonema bourbonense* | KC736621 | 0.007 | | 1.000 | 1.000 | 0.999 | 0.990 | 0.992 | 0.992 | 0.991 | 0.992 | 0.974 | 0.974 | 0.972 | 0.972 | 0.968 |
| **3** *Gomphonema bourbonense* | KC736620 | 0.007 | 0.000 | | 1.000 | 0.999 | 0.990 | 0.992 | 0.992 | 0.991 | 0.992 | 0.974 | 0.974 | 0.972 | 0.972 | 0.968 |
| **4** *Gomphonema bourbonense* | JN790281 | 0.007 | 0.000 | 0.000 | | 0.999 | 0.990 | 0.992 | 0.992 | 0.991 | 0.992 | 0.974 | 0.974 | 0.972 | 0.972 | 0.968 |
| **5** *Gomphonema bourbonense* | KT072957 | 0.009 | 0.001 | 0.001 | 0.001 | | 0.989 | 0.990 | 0.990 | 0.990 | 0.991 | 0.972 | 0.972 | 0.972 | 0.972 | 0.967 |
| **6** *Gomphonema pumilum* | KC736629 | 0.008 | 0.010 | 0.010 | 0.010 | 0.012 | | 0.992 | 0.992 | 0.992 | 0.992 | 0.973 | 0.973 | 0.972 | 0.972 | 0.967 |
| **7** *Gomphonema acuminatum* | JN790280 | 0.008 | 0.008 | 0.008 | 0.008 | 0.009 | 0.008 | | 1.000 | 0.999 | 0.999 | 0.978 | 0.978 | 0.977 | 0.977 | 0.969 |
| **8** *Gomphonema acuminatum* | AM502019 | 0.008 | 0.008 | 0.008 | 0.008 | 0.009 | 0.008 | 0.000 | | 0.999 | 0.999 | 0.978 | 0.978 | 0.977 | 0.977 | 0.969 |
| **9** *Gomphonema subclavatum* | KJ011666 | 0.009 | 0.009 | 0.009 | 0.009 | 0.010 | 0.009 | 0.001 | 0.001 | | 0.999 | 0.977 | 0.977 | 0.976 | 0.976 | 0.969 |
| **10** *Gomphonema truncatum* | AM501956 | 0.009 | 0.007 | 0.007 | 0.007 | 0.009 | 0.009 | 0.001 | 0.001 | 0.001 | | 0.977 | 0.977 | 0.976 | 0.976 | 0.970 |
| **11** *Gomphonema parvulum* | KC736625 | 0.024 | 0.026 | 0.026 | 0.026 | 0.027 | 0.027 | 0.022 | 0.022 | 0.022 | 0.022 | | 1.000 | 0.997 | 0.998 | 0.985 |
| **12** *Gomphonema parvulum* | KT072961 | 0.024 | 0.026 | 0.026 | 0.026 | 0.027 | 0.027 | 0.022 | 0.022 | 0.022 | 0.022 | 0.000 | | 0.997 | 0.998 | 0.985 |
| **13** *Gomphonema dichotomum* | KJ011655 | 0.025 | 0.027 | 0.027 | 0.027 | 0.028 | 0.028 | 0.023 | 0.023 | 0.024 | 0.024 | 0.003 | 0.003 | | 0.998 | 0.985 |
| **14** *Gomphonema affine* | MH997843 | 0.025 | 0.027 | 0.027 | 0.027 | 0.028 | 0.028 | 0.023 | 0.023 | 0.024 | 0.024 | 0.002 | 0.002 | 0.002 | | 0.987 |
| **15** *Gomphonema clevei* | KT072959 | 0.031 | 0.031 | 0.031 | 0.031 | 0.032 | 0.033 | 0.030 | 0.030 | 0.031 | 0.029 | 0.015 | 0.015 | 0.015 | 0.013 | |
| | | *p*-distance | | | | | | | | | | | | | | |

Accession; Gene Bank accession number.

**Table 7.** Similarity scores and genetic distances among 12 aligned sequences (596 bp) of *Gomphonema* species based on rbcL gene.

| Species | Accession | 1 | 2 | 3 | 4 | 5 | 6 | 7 | 8 | 9 | 10 | 11 | 12 |
|---|---|---|---|---|---|---|---|---|---|---|---|---|---|
| | | Similarity | | | | | | | | | | | |
| 1 *Gomphonema seminulum* | OP183210 | | 0.955 | 0.949 | 0.962 | 0.962 | 0.960 | 0.953 | 0.942 | 0.942 | 0.948 | 0.948 | 0.927 |
| 2 *Gomphonema bourbonense* | KC736595 | 0.044 | | 0.947 | 0.962 | 0.958 | 0.953 | 0.957 | 0.948 | 0.948 | 0.942 | 0.942 | 0.926 |
| 3 *Gomphonema pumilum* | KC736599 | 0.049 | 0.050 | | 0.957 | 0.957 | 0.957 | 0.957 | 0.946 | 0.946 | 0.944 | 0.944 | 0.933 |
| 4 *Gomphonema acuminatum* | LR742550 | 0.037 | 0.042 | 0.037 | | 0.997 | 0.990 | 0.983 | 0.953 | 0.953 | 0.953 | 0.953 | 0.933 |
| 5 *Gomphonema acuminatum* | LR742553 | 0.037 | 0.042 | 0.040 | 0.003 | | 0.993 | 0.986 | 0.953 | 0.953 | 0.953 | 0.953 | 0.933 |
| 6 *Gomphonema subclavatum* | LR742556 | 0.039 | 0.042 | 0.045 | 0.010 | 0.007 | | 0.986 | 0.950 | 0.950 | 0.950 | 0.950 | 0.929 |
| 7 *Gomphonema truncatum* | LR742547 | 0.045 | 0.042 | 0.042 | 0.017 | 0.013 | 0.013 | | 0.950 | 0.950 | 0.949 | 0.949 | 0.933 |
| 8 *Gomphonema parvulum* | KY661571 | 0.055 | 0.052 | 0.050 | 0.045 | 0.045 | 0.049 | 0.049 | | 1.000 | 0.985 | 0.985 | 0.964 |
| 9 *Gomphonema parvulum* | KY661554 | 0.055 | 0.052 | 0.050 | 0.045 | 0.045 | 0.049 | 0.049 | 0.000 | | 0.985 | 0.985 | 0.964 |
| 10 *Gomphonema affine* | AM710469 | 0.050 | 0.054 | 0.055 | 0.045 | 0.045 | 0.049 | 0.049 | 0.015 | 0.015 | | 1.000 | 0.964 |
| 11 *Gomphonema affine* | MK576042 | 0.050 | 0.054 | 0.055 | 0.045 | 0.045 | 0.049 | 0.049 | 0.015 | 0.015 | 0.000 | | 0.964 |
| 12 *Gomphonema clevei* | JQ354682 | 0.069 | 0.064 | 0.070 | 0.064 | 0.064 | 0.067 | 0.064 | 0.035 | 0.035 | 0.035 | 0.035 | |
| | | *p*-distance | | | | | | | | | | | |

Accession; Gene Bank accession number.

*3.5. Morphological Characteristics of Nitzschia inclinata sp. nov.*

Class Bacillariophyceae
Subclass Bacillariophycidae D.G.
Order Bacillariales
Family Bacillariaceae
Genus *Nitzschia*
*Nitzschia inclinata* E.A. Hwang and B.H. Kim (Figure 10, LM; Figure 11, SEM)

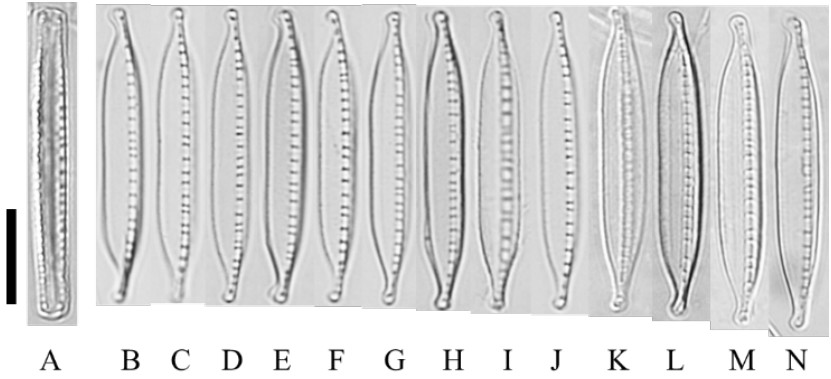

**Figure 10.** Light microscope photo of *Nitzschia inclinata* sp. nov. taken from holotype population (AG9004): (**A**) girdle view; (**B–N**) valve view; scale bar = 10 μm.

**Description:** Valves are narrow linear, and valve apices are capitate and slightly bent to one side. Length 32–34.5 μm, width 3.5–5 μm, 7–10 fibula in 10 μm, 50 striae in 10 μm, and 60–80 areolae in 10 μm. Striae are uniseriate, parallel, and not observed under LM. Areola on the valve is loculate in type and circular in shape. The terminal fissure is located at each valve apex with a broad sweeping curve, and a helictoglossa is formed inside the valve. Fibula is to one side of the valve face and is irregular in width. Conopeum is positioned on the external valve face along the fibula. On the face of the conopeum, circular areolae form one line, but for the valve apices, the number of areolae increases to

1–3. Raphe is eccentric from the valve face, centered on the conopeum. Support points connecting the valve face to the conopeum are observed. In girdle view, frustules are rectangular. Girdle bands with double rows of small punctuate are associated with a valve.

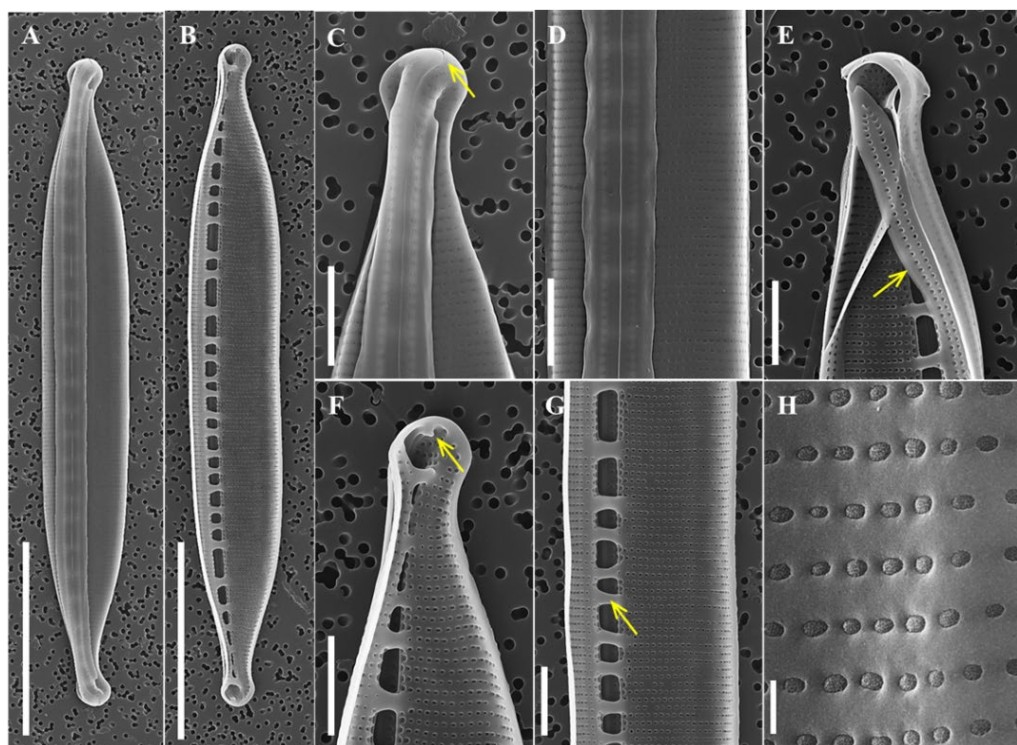

**Figure 11.** SEM of *Nitzschia inclinata* sp. nov. taken from holotype population (AG9004): (**A**,**B**) scale bar = 10 μm; (**C**–**G**) scale bar = 2 μm; (**H**) scale bar = 200 nm; (**A**) external view of whole valve; (**B**) internal view of whole valve; (**C**) external view of valve apex, terminal fissure (arrow); (**D**) external view of valve middle; (**E**) internal view of valve apex with girdle band (arrow); (**F**) internal view of valve apex, helictoglossa (arrow); (**G**) internal view of valve middle, fibulae (arrow); (**H**) external view of areolae.

**Remarks:** The morphology, including the ultrastructure, of *Nitzschia inclinata* was observed and compared with morphologically similar species belonging to the same genus (Table 8). The *N. dissipata* [41] valve is more lanceolate than *N. inclinata*.; *N. dissipata* formed subrostrate apices but *N. inclinata* has capitate apices. Another similar species, *N. oligotraphenta* [42], has linear lanceolate valves, but *N. inclinata* is narrow and linear in shape. A more obvious difference is that the valve apices of *N. oligotraphenta* are straight, whereas those of *N. inclinata* are slightly curved to one side of the valve. *N. sigmoidea* [43] is longer and larger than *N. inclinata*. In addition, in girdle view, *N. sigmoidea* is sigmoid, and *N. inclinata* is rectangular. Finally, *N. sigmoidea* fibula is along the margin of one side of the valve, but *N. inclinata* fibula is located on the valve face. The valve of *N. angularis* [44] is rhombic in shape, so the center of the valve is swollen and sharply narrowed toward the end of the apices. *N. inclinata* is clearly distinct from similar species in morphology and microstructure.

**Table 8.** Comparison of morphological characteristics for *Nitzschia inclinata* sp. nov. and closely related species.

| | *Nitzschia inclinata* | *N. dissipata* | *N. oligotraphenta* | *N. sigmoidea* | *N. angularis* |
|---|---|---|---|---|---|
| Length (µm) | 32–34.5 | 12.5–85 | 30–45 | 346–359 | 60–200 |
| Width (µm) | 3.5–5 | 3.5–7 | 3–4 | 9–13 | 6–15 |
| No. striae (/10 µm) | 50 | 39–50 | 46–48 | 24–26 | 31–32 |
| No. areolae (/10 µm) | 60–80 | 69 | n.d. | 40–50 | n.d. |
| No. fibulae (/10 µm) | 7–10 | 5–11.5 | 8.5–11.5 | 6–9 | 2.5–5 |
| Fibulae | One-sided valve, irregularly distributed | One side of valve, irregularly distributed | One-sided valve | One side of valve margin, irregularly distributed | Center of valve longitudinally striated |
| Striation | Parallel | Parallel | Parallel | Parallel | Parallel |
| Helictoglossa | Very apex of the valve (raphe ends internally formed) | n.d. | n.d. | Very apex of the valve (raphe ends internally formed) | Very apex of the valve |
| Valves | Narrow linear | Narrow lanceolate | Linear lanceolate | Linear | Rhomboidal |
| Valve apices | Capitate, bended | subrostrate | Distinctly capitulum | Tapering | n.d. |
| Raphe | Eccentric | Slightly eccentric | Eccentric | Highly eccentric, one side of valve margin | Almost central of valve face |
| Terminal fissure | Broad sweeping curve over the valve apex | Hook shape, sometimes widened or bifurcate | n.d. | Hook-shaped | Hook-shaped |
| Girdle view | Rectangular | n.d. | n.d. | Sigmoid | n.d. |
| Girdle band | Double rows of small punctuate | Double rows of small punctuate | n.d. | Double rows of small punctuate | n.d. |
| Reference | This study | [41,45–47] | [42,46,47] | [43] | [44] |

*Holotype:* permanent slides were deposited with the Korea Collection for Type Cultures KCTC under the deposit number AG9004.

*Habitat:* free-living on rocky substrates. Water quality: WT, 1.3 °C; DO, 14.6 mg/L; pH, 5.5; EC, 49 µS/cm; turbidity, 10.7 NTU. Major species (more than 5% of the total abundance) recorded from the same location: *Gomphoneis quadripunctata*, *Meridion constrictum*, *Achnanthidium minutissimum*, and *Achnanthes convergens*.

*Type locality:* Republic of Korea, Gyeonggi-do Province, Mount Yongma, 37°29′46″ N 127°16′55″ E, 3 January 2019.

*Molecular characterization:* SSU, rbcL accession no. The nucleotide sequences of the SSU rRNA and rbcL genes of the strain were deposited in GenBank (NCBI) under the accession numbers ON040638 and OP183211.

*Etymology:* "inclinata" is derived from the Latin "incliato", which means that the valve apices of *N. inclinata* are inclined to one side.

*3.6. Phylogenetic Characteristics of Nitzschia inclinata sp. nov.*

We performed phylogenetic analysis by selecting 20 aligned sequences of SSU rRNA and 18 aligned sequences of rbcL from the molecularly similar species to *N. inclinata*, and *Cyclotella meneghiniana* was used as an outgroup (Figures 12 and 13). In the SSU rRNA and rbcL phylogenetic trees, *N. inclinata* was located in the *Nitzschia* clade and was distinct from *Navicula* and *Amphora*. In SSU rRNA tree, *N. inclinata* was related to *N. dissipata* and *N. sigmoidea*, with strong support (ML bootstrap = 100%). The genetic distance between *N. inclinata* and these species was 0.016 for *N. dissipata* and 0.019 for *N. simoidea*, located within the same clade but distinct (Table 9). In the rbcL tree, *N. inclinata* was closely related to *N. sigmoidea* and *N. dissipata* and was the sister taxon of *N. sigmoidea* with 64% support. In rbcL-aligned sequences, the genetic distance between *N. inclinata* and *N. sigmodea* was 0.027, closer than to other species (Table 10). In the SSU rRNA and rbcL phylogenetic trees, *N. inclinata* was located on a branch close to *N. dissipata* and *N. sigmodea*, but its individual branch had high support values. Although they were close in genetic distance, they showed clear differences. Therefore, we are confident that *N. inclinata* is a new species with different molecular characteristics from similar species.

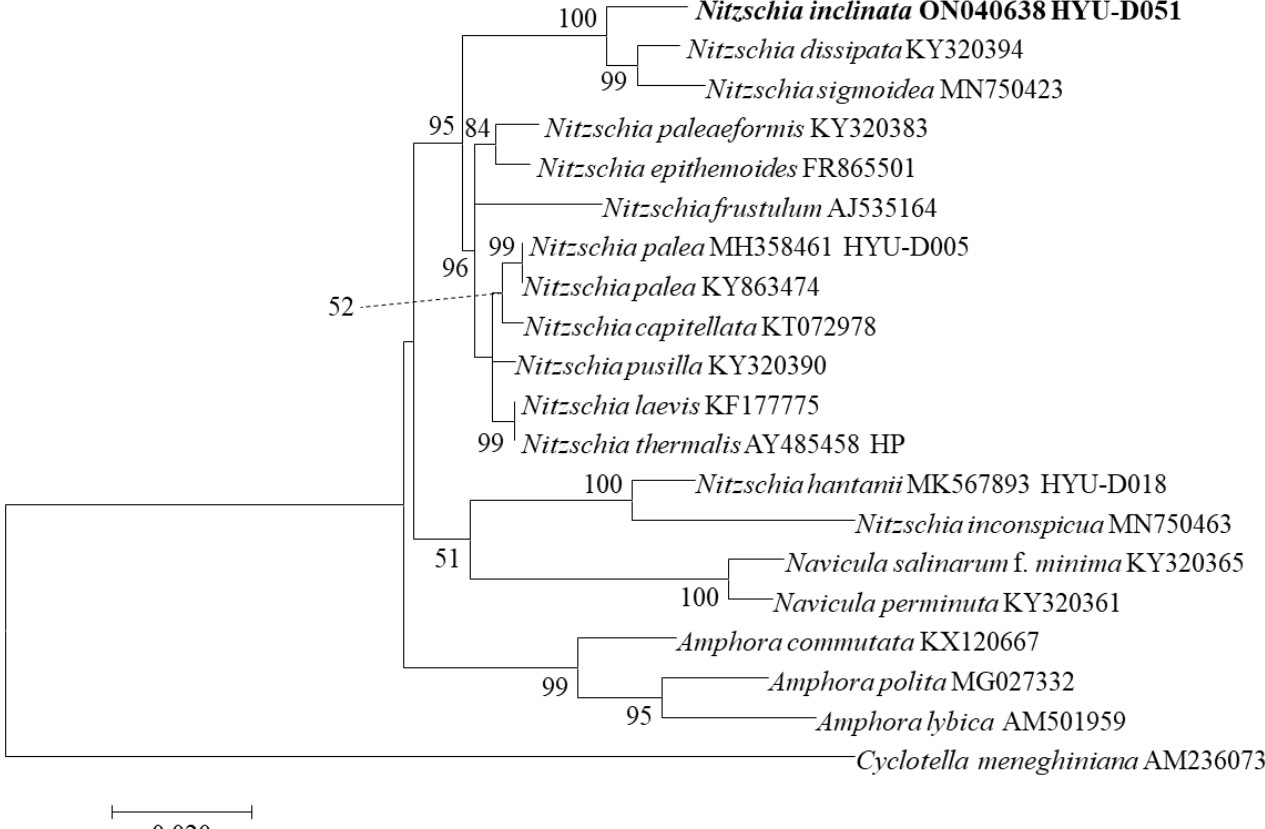

**Figure 12.** SSU rRNA phylogenetic tree by maximum likelihood method of *Nitzschia inclinata* sp. nov. and other species molecular position. The tree with the highest log likelihood (−4374.77) is shown. The percentages of trees in which the associated taxa clustered together are shown next to the branches. The analysis involved 20 nucleotide sequences and 1392 positions in the final data set. Cyclotella meneghiniana was used as the outgroup.

**Table 9.** Similarity scores and genetic distances among 14 aligned sequences (1392 bp) of *Nitzschia* species based on SSU rRNA gene.

| Species | Accession | 1 | 2 | 3 | 4 | 5 | 6 | 7 | 8 | 9 | 10 | 11 | 12 | 13 | 14 |
|---|---|---|---|---|---|---|---|---|---|---|---|---|---|---|---|
| | | Similarity | | | | | | | | | | | | | |
| 1 *Nitzschia inclinata* | ON040638 | | 0.984 | 0.981 | 0.978 | 0.976 | 0.976 | 0.975 | 0.975 | 0.974 | 0.975 | 0.975 | 0.963 | 0.953 | 0.937 |
| 2 *Nitzschia dissipata* | KY320394 | 0.016 | | 0.988 | 0.977 | 0.976 | 0.976 | 0.974 | 0.974 | 0.976 | 0.976 | 0.974 | 0.964 | 0.958 | 0.943 |
| 3 *Nitzschia sigmoidea* | MN750423 | 0.019 | 0.012 | | 0.975 | 0.972 | 0.972 | 0.974 | 0.974 | 0.971 | 0.972 | 0.970 | 0.959 | 0.953 | 0.938 |
| 4 *Nitzschia pusilla* | KY320390 | 0.022 | 0.022 | 0.025 | | 0.993 | 0.993 | 0.994 | 0.994 | 0.989 | 0.989 | 0.993 | 0.981 | 0.960 | 0.948 |
| 5 *Nitzschia palea* | MH358461 | 0.024 | 0.024 | 0.027 | 0.006 | | 1.000 | 0.995 | 0.995 | 0.991 | 0.990 | 0.995 | 0.982 | 0.962 | 0.951 |
| 6 *Nitzschia palea* | KY863474 | 0.024 | 0.024 | 0.027 | 0.006 | 0.000 | | 0.995 | 0.995 | 0.991 | 0.990 | 0.995 | 0.982 | 0.962 | 0.951 |
| 7 *Nitzschia laevis* | KF177775 | 0.025 | 0.025 | 0.026 | 0.006 | 0.005 | 0.005 | | 1.000 | 0.990 | 0.991 | 0.993 | 0.982 | 0.960 | 0.948 |
| 8 *Nitzschia thermalis* | AY485458 | 0.025 | 0.025 | 0.026 | 0.006 | 0.005 | 0.005 | 0.000 | | 0.990 | 0.991 | 0.993 | 0.982 | 0.960 | 0.948 |
| 9 *Nitzschia epithemoides* | FR865501 | 0.025 | 0.024 | 0.028 | 0.011 | 0.009 | 0.009 | 0.010 | 0.010 | | 0.993 | 0.990 | 0.979 | 0.958 | 0.949 |
| 10 *Nitzschia paleaeformis* | KY320383 | 0.025 | 0.024 | 0.027 | 0.011 | 0.010 | 0.010 | 0.009 | 0.009 | 0.007 | | 0.988 | 0.979 | 0.961 | 0.948 |
| 11 *Nitzschia capitellata* | KT072978 | 0.025 | 0.025 | 0.029 | 0.006 | 0.005 | 0.005 | 0.006 | 0.006 | 0.010 | 0.012 | | 0.980 | 0.961 | 0.948 |
| 12 *Nitzschia frustulum* | AJ535164 | 0.038 | 0.037 | 0.041 | 0.020 | 0.019 | 0.019 | 0.019 | 0.019 | 0.022 | 0.022 | 0.021 | | 0.949 | 0.936 |
| 13 *Nitzschia hantanii* | MK567893 | 0.045 | 0.041 | 0.045 | 0.039 | 0.037 | 0.037 | 0.039 | 0.039 | 0.040 | 0.038 | 0.038 | 0.051 | | 0.968 |
| 14 *Nitzschia inconspicua* | MN750463 | 0.061 | 0.055 | 0.059 | 0.051 | 0.048 | 0.048 | 0.051 | 0.051 | 0.049 | 0.050 | 0.050 | 0.063 | 0.031 | |
| | | *p*-distance | | | | | | | | | | | | | |

Accession: GenBank accession number.

**Table 10.** Similarity scores and genetic distances among 12 aligned sequences (557 bp) of *Nitzschia* species based on rbcL gene.

| Species | Accession | 1 | 2 | 3 | 4 | 5 | 6 | 7 | 8 | 9 | 10 | 11 | 12 |
|---|---|---|---|---|---|---|---|---|---|---|---|---|---|
| | | | | | | Similarity | | | | | | | |
| 1 *Nitzschia inclinata* | OP183211 | | 0.956 | 0.973 | 0.967 | 0.890 | 0.886 | 0.882 | 0.882 | 0.900 | 0.886 | 0.884 | 0.873 |
| 2 *Nitzschia dissipata* | KY320332 | 0.043 | | 0.948 | 0.956 | 0.867 | 0.871 | 0.871 | 0.871 | 0.900 | 0.873 | 0.871 | 0.860 |
| 3 *Nitzschia sigmoidea* | FN557033 | 0.027 | 0.050 | | 0.965 | 0.892 | 0.882 | 0.877 | 0.877 | 0.902 | 0.886 | 0.879 | 0.875 |
| 4 *Nitzschia* cf. *sigmoidea* | KM999113 | 0.032 | 0.043 | 0.034 | | 0.890 | 0.890 | 0.881 | 0.881 | 0.912 | 0.896 | 0.888 | 0.871 |
| 5 *Nitzschia capitellata* | KT072924 | 0.102 | 0.122 | 0.101 | 0.102 | | 0.976 | 0.965 | 0.965 | 0.912 | 0.926 | 0.928 | 0.916 |
| 6 *Nitzschia palea* | KC736609 | 0.106 | 0.118 | 0.110 | 0.102 | 0.023 | | 0.989 | 0.989 | 0.914 | 0.912 | 0.925 | 0.914 |
| 7 *Nitzschia palea* | KF959639 | 0.110 | 0.118 | 0.113 | 0.110 | 0.034 | 0.011 | | 1.000 | 0.906 | 0.904 | 0.917 | 0.906 |
| 8 *Nitzschia palea* | KJ542464 | 0.110 | 0.118 | 0.113 | 0.110 | 0.034 | 0.011 | 0.000 | | 0.906 | 0.904 | 0.917 | 0.906 |
| 9 *Nitzschia paleaeformis* | KY320322 | 0.093 | 0.093 | 0.092 | 0.083 | 0.083 | 0.081 | 0.088 | 0.088 | | 0.930 | 0.940 | 0.928 |
| 10 *Nitzschia hantanii* | MK576040 | 0.106 | 0.117 | 0.106 | 0.097 | 0.070 | 0.083 | 0.090 | 0.090 | 0.066 | | 0.961 | 0.950 |
| 11 *Nitzschia inconspicua* | KC736607 | 0.108 | 0.118 | 0.111 | 0.104 | 0.068 | 0.072 | 0.079 | 0.079 | 0.057 | 0.038 | | 0.969 |
| 12 *Nitzschia frustulum* | HF675070 | 0.117 | 0.127 | 0.115 | 0.118 | 0.079 | 0.081 | 0.088 | 0.088 | 0.068 | 0.048 | 0.031 | |
| | | | | | | *p*-distance | | | | | | | |

Accession; Gene Bank accession number.

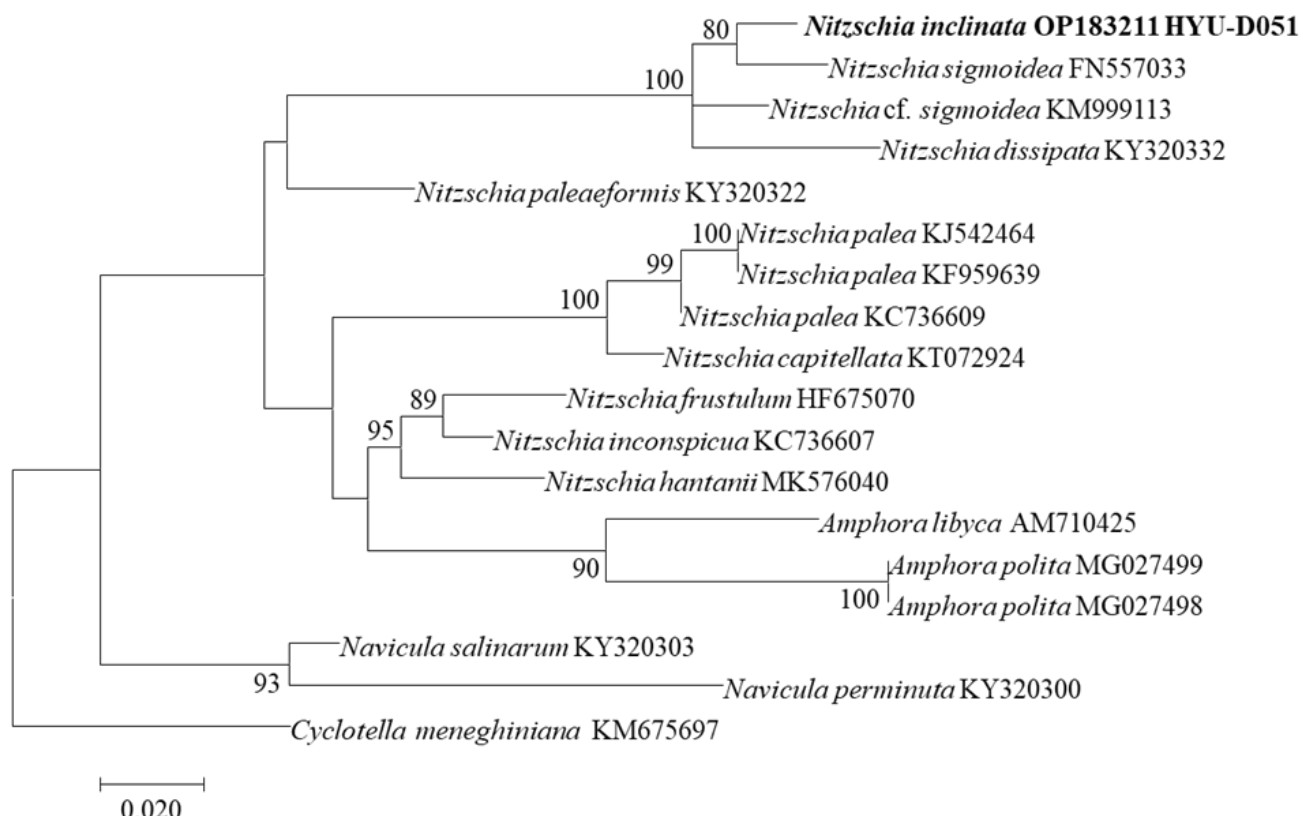

**Figure 13.** rbcL phylogenetic tree by maximum likelihood method of *Nitzschia inclinata* sp. nov. and other species molecular position. The tree with the highest log likelihood (−2490.51) is shown. The percentages of trees in which the associated taxa clustered together are shown next to the branches. The analysis involved 18 nucleotide sequences and 557 positions in the final data set. Cyclotella meneghiniana was used as the outgroup.

**Author Contributions:** E.-A.H. and B.-H.K. conceived the research; E.-A.H., H.-K.K., I.-H.C., and B.-H.K. performed fieldwork; E.-A.H., B.-H.K. and C.Y. analyzed the data; E.-A.H. and B.-H.K. wrote and edited the manuscript. All authors have read and agreed to the published version of the manuscript.

**Funding:** This work was supported by a grant from the Nakdonggang National Institute of Biological Resources (NNIBR) funded by the Ministry of Environment (MOE) of the Republic of Korea (NNIBR202201205).

**Institutional Review Board Statement:** Not applicable.

**Data Availability Statement:** Not applicable.

**Acknowledgments:** We thank anonymous reviewers for their valuable and constructive comments, and special thanks to Kim, Y.-H. for kind assistance.

**Conflicts of Interest:** The authors report no potential conflict of interest.

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
