# Peer review of "Morphological and Molecular Studies of Three New Diatom Species from Mountain Streams in South Korea"

_diversity, doi:10.3390/d14100790_

Round 1

Reviewer 1 Report

Well-written paper on three new diatom species from mountain streams in South Korea, with both morphological as well as molecular description.

However, before the manuscript can be accepted for publication, an important correction on the holotype is needed beside some minor revisions.

The holotype can be only one slide, made from the type material. If multiple slides are made from the same type material, one slide must be chosen as the holotype, the others are the isotypes. The authorities should not be written in italics, and adding the authorities to the names of the genera would be an added value. For the new Hannea the apical pore field is not mentioned in the description. The captions of the LM and SEM figures should indicate which materials were used. The authors have to verify the correct spelling and translation of the Latin words from which the epithets are derived.  

The description of Gomphonema seminulum can be more extended, e.g. intern central raphe endings forming a right angle. The comparison with the striation of G. pumilum on line 273 “striae are slight radiate to strongly radiate towards the apices but G. seminulum is almost parallel in central area and slightly radiate in apices” must be checked. The valve depicted in Fig. 7 B has strongly radiate striae near the foot pole. Correct this also in the description. Information on the distribution of the closely related taxa is lacking. Although no information is available on Genbank, are there morphological closely related taxa of Gomphonema seminulum  from tropical regions, especially from Africa, Gomphonema clevei for example?

Registration of the three new species to Phycobank is recommended.

Although I am not a native English speaker, I noticed that some improvements should be made in the English of the text. Furthermore, the exponents must be written correctly in superscript, such as be cm² and cm³ and a space is needed between the number and the unit (25 cm²). Also in other places in the text sometimes a point is missing (e.g. on line 92…Table 1.PCR…).

In the introduction the authors mentioned that there are over 10 000 diatom species described, with the reference of Mann (1999). This paper dates back to more than 20 years. Meanwhile many new diatoms are described worldwide and their number increased to an estimation (between 12 000 and 30 000) (e.g., Guiry 2012, Mann & Vanormelingen 2013, Malviya et al. 2016, …). Also refer by the environmental conditions on line 25 to salinity/conductivity and pH in addition to water temperature and nutrients.

Check the list with the references carefully. For example for reference 5 a point is needed after the title of the chapter, abbreviations of the first names of the editors are incorrect and the page numbers are lacking. (Title of the chapter. In Book Title, 2nd ed.; Editor 1, A., Editor 2, B., Eds.; Publisher: Publisher Location, Country, Year; Volume 3, pp. xx–xxx.)

Some specific remarks:

Line 60. Add the type of inverted microscope used.

Line 70-75. “Morphological analysis….per sample”. These sentences belong to a new paragraph: “Diatom preparation for permanent slides”.

Line 78. Add the refraction index of the Mountain media used

Line 84. Add the type of coating device and the type of the SEM.

Line 194 Typing error in Ulnaria

Author Response

Response to Reviewer 1 Comments

Point 1: The holotype can be only one slide, made from the type material. If multiple slides are made from the same type material, one slide must be chosen as the holotype, the others are the isotypes.

Answer: Thank you for your comment. We specified the holotype and the isotype among the slides.

Point 2: The authorities should not be written in italics, and adding the authorities to the names of the genera would be an added value.

Answer: Thank you for your comment. We revised it.

Point 3: For the new Hannea the apical pore field is not mentioned in the description. The captions of the LM and SEM figures should indicate which materials were used. The authors have to verify the correct spelling and translation of the Latin words from which the epithets are derived.  

Answer: Thank you for your comment. Added description to vertex pore field in result 3.1. In addition, the material numbers used for LM and SEM figures are indicated in the caption.

Point 4: The description of Gomphonema seminulum can be more extended, e.g. intern central raphe endings forming a right angle. The comparison with the striation of G. pumilum on line 273 “striae are slight radiate to strongly radiate towards the apices but G. seminulum is almost parallel in central area and slightly radiate in apices” must be checked. The valve depicted in Fig. 7 B has strongly radiate striae near the foot pole.

Answer: Thank you for your comment. We performed detailed observations of the Gomphonema seminulum. The striae near the headpole is weakly radiated, but it is relatively strongly radiated near the footpole. The result of 3.3 and table 5 have been corrected.

Point 5: Correct this also in the description. Information on the distribution of the closely related taxa is lacking. Although no information is available on Genbank, are there morphological closely related taxa of Gomphonema seminulum from tropical regions, especially from Africa, Gomphonema clevei for example?

Answer: The phylogenetic tree contains closely related taxa (G. bourbonense, G. pumilum). G. clevei is included in the tree, and similarities, and p-distances indicate relatively large distances from G. seminulum. Therefore, it is considered to have been sufficiently explained.

Point 6: Registration of the three new species to Phycobank is recommended.

Response 6: Thank you for your recommend. We plan to contact the Phycobank staff to register.

Point 7: Although I am not a native English speaker, I noticed that some improvements should be made in the English of the text. Furthermore, the exponents must be written correctly in superscript, such as be cm² and cm³ and a space is needed between the number and the unit (25 cm²). Also in other places in the text sometimes a point is missing (e.g. on line 92…Table 1.PCR…).

Answer: Thank you for your comment. We fixed error.

Point 8: In the introduction the authors mentioned that there are over 10 000 diatom species described, with the reference of Mann (1999). This paper dates back to more than 20 years. Meanwhile many new diatoms are described worldwide and their number increased to an estimation (between 12 000 and 30 000) (e.g., Guiry 2012, Mann & Vanormelingen 2013, Malviya et al. 2016, …). Also refer by the environmental conditions on line 25 to salinity/conductivity and pH in addition to water temperature and nutrients.

Answer: Thank you for your comment. With reference to Mann & Vanormelingen (2013) paper, we wrote that more than 30,000 species of diatoms are estimated. And referring to the papers of Kovács et al.(2006), Pestryakova et al. (2018) and Kim et al. (2019) on line 25, we explained that the diatom is regulated by various environmental factors (pH, conductivity, salinity).

(1. Mann, D.G.; Vanormelingen, P. An inordinate fondness? The number, distributions, and origins of diatom species. J. Eukaryot. Microbiol. 2013, 60, 414-420.

  1. Kovács, C.; Kahlert, M.; Padisák, J. Benthic diatom communities along pH and TP gradients in Hungarian and Swedish streams. J. Appl. Phycol. 2006 18, 105-117.
  2. Pestryakova, L.A.; Herzschuh, U.; Gorodnichev, R.; Wetterich, S. The sensitivity of diatom taxa from Yakutian lakes (north-eastern Siberia) to electrical conductivity and other environmental variables. Polar Res. 2018, 37.
  3. Kim, H.K.; Cho, I.H.; Hwang, E.A.; Kim, Y.J.; Kim, B.H. Benthic diatom communities in Korean estuaries: species appearances in relation to environmental variables. Int. J. Environ. Health Res. 2019, 16.)

Point 9: Check the list with the references carefully. For example for reference 5 a point is needed after the title of the chapter, abbreviations of the first names of the editors are incorrect and the page numbers are lacking. (Title of the chapter. In Book Title, 2nd ed.; Editor 1, A., Editor 2, B., Eds.; Publisher: Publisher Location, Country, Year; Volume 3, pp. xx–xxx.)

Answer: Thank you for your comment. We double-checked the references form to get it right.

Some specific remarks:

Point 10: Line 60. Add the type of inverted microscope used.

Answer: Thank you for your comment. We added type.

Point 11: Line 70-75. “Morphological analysis….per sample”. These sentences belong to a new paragraph: “Diatom preparation for permanent slides”.

Answer: Thank you for your comment. We added the new paragraph.

Point 12: Line 78. Add the refraction index of the Mountain media used.

Answer: Thank you for your comment. We added the refraction index.

Point 13: Line 84. Add the type of coating device and the type of the SEM.

Answer: Thank you for your comment. We added it.

Point 14: Line 194 Typing error in Ulnaria

Answer: Thank you for your comment. We fixed typing error.

Reviewer 2 Report

The authors describe three new species of diatoms. The morphological and phylogenetic analysis methods are well designed and therefore the results are consistent.
Any doubts:
In methods, the authors only use the term diatom. One must differentiate between "species" and "cell" or "individual". For example, how many isolated cells are used to start the culture of a certain species before performing the phylogenetic analysis methods? The morphological comparisons are well argued, although it would have been very illustrative to show images of the most similar species highlighting the differences.
In fig. 2 there are only images of two sizes of H. librata, large or small: it seems logical to expect continuous variations in size.
Phylogenetic analysis of G. semilunum allows a direct comparison with very similar species (G. pumilum, G. bourbonensis).
However, the phylogenetic analysis of H. librata does not include any other species in the genus Hannaea even though H. librata is morphologically compared to H. pamirensis and H. hattoriana.
The phylogenetic analysis of N. inclinata includes other diatoms of the genus Nitzschia but not a very similar species such as N. oligotraphenta. These doubts do not require redoing the paper, although it would be convenient to explain the absence of phylogenetic data for some species, in addition to clarifying some methodological aspect. In short, this paper can be published with minor changes.

Author Response

Response to Reviewer 2 Comments

The authors describe three new species of diatoms. The morphological and phylogenetic analysis methods are well designed and therefore the results are consistent.

Any doubts:

Point 1: In methods, the authors only use the term diatom. One must differentiate between "species" and "cell" or "individual". For example, how many isolated cells are used to start the culture of a certain species before performing the phylogenetic analysis methods? The morphological comparisons are well argued, although it would have been very illustrative to show images of the most similar species highlighting the differences.

Answer: Thank you for your comment. In methods, "diatom" has been replaced with a word that fits the meaning ("species" and "cell" or "individual").

Point 2: In fig. 2 there are only images of two sizes of H. librata, large or small: it seems logical to expect continuous variations in size.

Answer: Thank you for your comment. Although we isolated two size of Hannaea librata, We think that data of more various sizes should be supplemented in further study.

Point 3: Phylogenetic analysis of G. semilunum allows a direct comparison with very similar species (G. pumilum, G. bourbonensis). However, the phylogenetic analysis of H. librata does not include any other species in the genus Hannaea even though H. librata is morphologically compared to H. pamirensis and H. hattoriana.

Answer: Thank you for your comment. Although Hannaea has distinct morphological characteristics that are different from similar genus and species, it is clearly distinguished, but several species are not distributed worldwide, and molecular characteristics have not been studied. In this study, the molecular location of the genus Hannaea was shown for the first time, and it can be used as basic data for further studies. After that, We think that research through molecular analysis of Hannaea species should be done.

Point 4: The phylogenetic analysis of N. inclinata includes other diatoms of the genus Nitzschia but not a very similar species such as N. oligotraphenta. These doubts do not require redoing the paper, although it would be convenient to explain the absence of phylogenetic data for some species, in addition to clarifying some methodological aspect. In short, this paper can be published with minor changes.  

Answer: Thank you for your comment. N. inclinata is clearly distinguished morphologically from similar species, and clearly distinguished molecularly from several morphologically similar species. On the other hand, molecular analysis data of diatoms are inferior to those of morphological analysis. Therefore, although molecular data of all species cannot be compared, N. inclinata is distinguished from other species through morphological and similar species analysis. However, further studies require a molecular comparison of N. oligotraphenta.